# Synthesis and Hydrogelation of Star-Shaped Graft Copolypetides with Asymmetric Topology

**DOI:** 10.3390/gels8060366

**Published:** 2022-06-09

**Authors:** Thi Ha My Phan, Yu-Hsun Yang, Yi-Jen Tsai, Fang-Yu Chung, Tooru Ooya, Shiho Kawasaki, Jeng-Shiung Jan

**Affiliations:** 1Department of Chemical Engineering, National Cheng Kung University, Tainan 70101, Taiwan; n36087113@gs.ncku.edu.tw (T.H.M.P.); e34074093@gs.ncku.edu.tw (Y.-H.Y.); n36094704@gs.ncku.edu.tw (Y.-J.T.); f54081036@gs.ncku.edu.tw (F.-Y.C.); 2Graduate School of Engineering, Kobe University, Kobe 657-8501, Japan; ooya@tiger.kobe-u.ac.jp (T.O.); shihoshi@stu.kobe-u.ac.jp (S.K.); 3Center of Advanced Medical Engineering Research & Development (CAMED), Kobe University, Kobe 657-8501, Japan; 4Hierarchical Green-Energy Materials (Hi-GEM) Research Center, National Cheng Kung University, Tainan 70101, Taiwan

**Keywords:** hydrogel, star-shaped polymer, polypeptide, topology

## Abstract

To study the self-assembly and hydrogel formation of the star-shaped graft copolypeptides with asymmetric topology, star-shaped poly(_L_-lysine) with various arm numbers were synthesized by using asymmetric polyglycerol dendrimers (PGDs) as the initiators and 1,1,3,3-tetramethylguanidine (TMG) as an activator for OH groups, followed by deprotection and grafting with indole or phenyl group on the side chain. The packing of the grafting moiety via non-covalent interactions not only facilitated the polypeptide segments to adopt more ordered conformations but also triggered the spontaneous hydrogelation. The hydrogelation ability was found to be correlated with polypeptide composition and topology. The star-shaped polypeptides with asymmetric topology exhibited poorer hydrogelation ability than those with symmetric topology due to the less efficient packing of the grafted moiety. The star-shaped polypeptides grafted with indole group on the side chain exhibited better hydrogelation ability than those grafted with phenyl group with the same arm number. This report demonstrated that the grafted moiety and polypeptide topology possessed the potential ability to modulate the polypeptide hydrogelation and hydrogel characteristics.

## 1. Introduction

Hydrogels are a type of biomaterials possessing the ability to contain a large amount of water inside their three-dimensional network, which makes their special properties such as living cells mimetic, biodegradable, function and conformation adjustable [1,2,3,4,5]. Moreover, with tunable mechanical properties, hydrogels have drawn significant attention to investigate their applications in biomedical fields such as wound dressing, drug delivery systems, and tissue engineering [6,7,8]. Hydrogel was defined first time by Bemmelen in 1894, and its first application was reported in biomedical fields by Wichterle and Lim in 1960 [9,10,11]. Amongst all types of hydrogels, peptide-based hydrogels have self-assembly ability for hydrogel formation through hydrogen bonding, aromatic (cation-π, π-π), and hydrophobic/hydrophilic interactions [12,13,14,15]. Additionally, peptide segments can adopt secondary conformations (random coil, β-sheet/β-turn) inside their gel network to enhance the mechanical properties of hydrogels [16,17,18,19,20]. As a type of synthetic hydrogel, polypeptides can modify the characteristics of hydrogels by varying the functional groups on the side chain or the chain composition on the backbone [21,22,23].

Traditionally, star-shaped polypeptides were synthesized by using dendrimers bearing many amino groups with symmetric topology as the initiators [24,25]. In 2016, the Heise group reported the synthesis of 8-armed block copolypeptides using a polypropylene imine dendrimer as the core of the polymers, and the hydrogelation of those samples was good with critical gelation concentrations (CGCs) lower than 2.0 wt% [25]. Recently, the polyol initiators were used with the promotion of 1,1,3,3-tetramethylguanidine (TMG) as an activator for hydroxyl (OH) groups to favor the ring-opening polymerization (ROP) of *N*-carboxyanhydride (NCA) reaction [21,22,26,27]. Most of the previous studies investigated the hydrogelation of the star-shaped block/graft polypeptides with symmetric structures [22,23]. In 2018 and 2019, our group reported that the star-shaped graft copolypeptides using the symmetric polyol initiators exhibited good hydrogelation ability in deionized water [23,28].

In this study, we synthesized the star-shaped polypeptides grafted with indole and phenyl groups, using polyglycerol dendrimers of generations 1, 2, and 3 (PGDs G1, G2, and G3) [27,29] as the asymmetric initiators to investigate the effect of grafted moiety, arm number, and polypeptide topology on the hydrogelation of the star-shaped graft copolypeptides. One previous paper has used PGD G2 as an initiator for the synthesis of the block copolypeptides and studied the hydrogelation of these polypeptides [21]. The PGD G1 and G3 have never been used as initiators for the synthesis of polypeptides. Herein, this study reported the synthesis and hydrogelation of asymmetric, star-shaped graft copolypeptides with 6, 12, and 24 arms using PGD G1, G2, and G3 as initiators, respectively. Moreover, indole and phenyl groups were employed as the grafting moieties for the polypeptides. The as-prepared polypeptides were characterized by a variety of analytical instruments to confirm the successful synthesis. Furthermore, these samples were analyzed to characterize their polypeptide chain conformation, assembled nano-/micro-structure in the hydrogel network, and hydrogel mechanical strength. Recovery behavior was also conducted to demonstrate their injectable ability.

## 2. Results and Discussion

### 2.1. Star-Shaped Graft Copolypeptide Synthesis and Characterization

As shown in Figure 1, the 6-armed poly(Z-_L_-lysine), 12-armed poly(Z-_L_-lysine), and 24-armed poly(Z-_L_-lysine) (6s-PZLL, 12s-PZLL, 24s-PZLL) homopolypeptides were synthesized by ROP of ZLL NCA with PGD G1, G2, and G3 as an asymmetric initiator, respectively, using TMG to activate the OH groups. The homopolypeptides were dissolved in trifluoroacetic acid-*d*_1_ (TFA-*d*_1_) and *N*, *N*-dimethylformamide (DMF) for proton nuclear magnetic resonance (^1^H NMR) and gel permeation chromatography–light scattering (GPC-LS) analyses, respectively, and the results for homopolypeptide characterization were summarized in Appendix A, Figure 1 and Appendix A. Based on ^1^H NMR data (Appendix A and Figure 1a), the degree of polymerization (DP) was calculated from the ratio of integral areas of d protons on the phenyl group of ZLL block to b protons on the initiator. The integrated values of other protons on ZLL could be used to cross-check the results. Moreover, the calculation of DPs and number-averaged molecular weights (M_n_) derived from the GPC-LS analysis gave good agreement with those from ^1^H NMR analysis (Appendix A). As shown in Appendix A, the proton b on the initiators at around 4.0–4.1 ppm demonstrated the successful synthesis of the star-shaped homopolypeptides by the activating ability of TMG on OH groups of the PGD initiators. This result is consistent with those from ^1^H–^13^C two-dimensional Heteronuclear Single Quantum Correlation (^1^H–^13^C 2D HSQC) analysis in a previous study, which reported that the proton b on the initiator was shifted from 3.82 ppm to 4.1 ppm after blocked with PLL [21]. After removing the Z groups on ZLL segments by using HBr, the star-shaped poly(_L_-lysine) (s-PLL) samples were dialyzed against deionized water (DIW) and obtained by freeze-drying. The s-PLL samples dissolved in deuterated water (D_2_O) were analyzed by ^1^H NMR. For an example of 12s-PLL in Figure 1b’, it is obvious that the Z groups were successfully removed with the remaining percentage lower than 5.0%, as reported in the literature [21,22,27].

The s-PLL polypeptides were conjugated with indolyl and phenyl groups by reacting with 3-indoleacetic acid and benzoic acid, respectively, using 1-ethyl-3-(-3-dimethylaminopropyl) carbodiimide/*N*-hydroxysuccinimide (EDC/NHS) as activators (an example of 6s-PLL-*g*-Indo as shown in Figure 1a). After dialysis and lyophilization, the samples were dissolved in D_2_O for ^1^H NMR analysis. The integrated ratio of the protons on the graft moiety to the α_1_ proton on the PLL block was calculated to determine the grafting ratio and grafting efficiency (Figure 1c’,d’). In Table 1, the grafting efficiency was in the range 26–43%, which is much lower than those in a reported study (75–83%) for 6-armed graft copolypeptides using dipentaerythritol as an initiator [23]. It could be possibly explained by the steric hindrance of the higher number of arms and asymmetric topology.

### 2.2. Critical Gelation Concentrations (CGCs) and Hydrogelation of Star-Shaped Graft Copolypeptides

The star-shaped graft copolypeptides were homogeneously dissolved in DIW by vortex mixing and sonication before the vial inversion approach to determine whether their CGCs were lower than 10.0 wt%. It is obvious that the 12-armed and 24-armed graft copolypeptides have good hydrogelation ability, with the CGC values ranging from 2.9 to 6.3 wt%, whilst the 6-armed ones cannot form hydrogel at a concentration lower than 10.0 wt% (Table 1). Moreover, the 24-armed poly(_L_-lysine)_12_-*graft*-indole_0.11_ (24s-PLL_12_-*g*-Indo_0.11_) (2.9 wt%) and 24-armed poly(_L_-lysine)_12_-*graft*-phenyl_0.13_ (24s-PLL_12_-*g*-Phenyl_0.13_) (3.6 wt%) exhibited lower CGCs than the 12s-PLL_12_-*g*-Indo_0.10_ (5.85 wt%) and 12s-PLL_12_-*g*-Phenyl_0.08_ (6.3 wt%). It revealed that the star-shaped polypeptides with comparable grafting ratio exhibited better hydrogelation ability upon the increase in the arm number. Moreover, the star-shaped polypeptides grafted with indole group exhibited better hydrogelation ability than those grafted with phenyl group. The results are consistent with those reported in the previous studies [21,22,23,27]. It can be explained that higher number of arms gave more branching chains in the polypeptide structure. We previously reported that 6-armed poly(_L_-lysine)_30_-*graft*-indole_0.30_ (6s-PLL_30_-*g*-Indo_0.30_) synthesized by using dipentaerythritol as the initiator possessed an excellent hydrogelation with the CGC of 0.75 wt%, whilst this study showed that 6s-PLL_32_-*g*-Indo_0.08_ using PGD G1 with asymmetric topology did not form hydrogel below 10.0 wt% [23]. It demonstrated that polymer topology has a strong effect on the hydrogelation of the star-shaped polypeptides and the symmetric topology could render the star-shaped polypeptides exhibiting a better gel-forming behavior via efficient packing. With the same arm number, the star-shaped polypeptides conjugated with indole group showed better hydrogelation ability than the ones conjugated with phenyl group. The hydrogen bonding, hydrophobic, and aromatic (including cation-π, π-π) interactions of the indole group could establish efficient facilitation for the hydrogelation of the star-shaped polypeptides. Although the grafted phenyl group cannot efficiently facilitate the formation of hydrogels as good as the grafted indole group, the hydrophobic and aromatic interactions of the phenyl group still could assist the hydrogelation of the corresponding polypeptides with low CGCs (6.3 wt% for 12s-PLL_12_-*g*-Phenyl_0.08_ and 3.6 wt% for 24s-PLL_12_-*g*-Phenyl_0.13_). Using PGD G2 as a polyol initiator, a previous study reported that 12-armed-poly(_L_-lysine)_16_-*block*-poly(_L_-phenylalanine)_5_ (12s-PLL_16_-*b*-PLF_5_) exhibited a CGC of 3.0 wt% [21], which is lower than that of 12s-PLL_12_-*g*-Phenyl_0.08_ (6.3 wt%). It suggested that the star-shaped block polypeptides have a better ability to form hydrogel than the star-shaped graft counterparts. The hydrophobic segment on the star-shaped block polypeptides could exhibit higher efficiency in packing in an aqueous environment than the hydrophobic group on the star-shaped graft counterparts. Amongst all samples, the 24s-PLL_12_-*g*-Indo_0.11_ exhibited the best hydrogelation ability, evidenced by its lowest CGC of 2.9 wt%. The results demonstrate that the functional groups and symmetry have a critical effect on the hydrogel formation of the star-shaped graft polypeptides.

### 2.3. Secondary Structure of Star-Shaped Graft Copolypeptides

Circular dichroism (CD) measurement was employed to determine the adoption of the secondary conformation for the polypeptide samples under neutral environments (Appendix A). It was obvious that a negative dominant peak was shown at 190–200 nm in the CD spectra of all polypeptide samples, which indicated the existence of random coil conformation. Moreover, a curved peak existing at a wavelength range of 215–225 nm was representative of β-motif conformation. The CD results were fitted to compute the percentage of polypeptide secondary conformation by using BeStSel website as shown in Table 1. The major conformations adopted by the star-shaped graft copolypeptides were random coil and β-motifs (β-sheet and β-turn). It was renowned that the random coil conformation was originated from the PLL segment at a neutral environment [21,22,29,30]. Upon grafting the functional group, the PLL segment would undergo coil-to-sheet conformational transition, due to the multiple interactions between the functional group. The results are in a good agreement with those reported in the previous studies [31,32,33]. As shown in Table 1, the percentage of β-sheet/β-turn conformations (50.0–60.0%) was much higher than the molar ratio of the functional groups grafted onto the star-shaped polypeptides. It demonstrated that the confinement of the PLL segment exerted by the packing of the grafted moiety would facilitate the polypeptide chains to adopt more ordered conformation (that is, β-sheet/β-turn) as well as hydrogelation [21,22,27]. It can be seen that the polypeptides grafted with indole group adopted slightly higher β-motif conformation than these grafted with phenyl group with the same arm number. The results exhibited that β-motif structures could enhance the hydrogelation in the network of the star-shaped graft polypeptides.

### 2.4. Self-Assembled Nanostructure and Gel Morphology of Star-Shaped Graft Copolypeptides

The macrostructures of the star-shaped graft copolypeptides were observed by field emission scanning electron microscopy (FE-SEM) analysis. The polypeptides were dissolved in DIW for gel formation and lyophilized to obtain the samples which possessed the native morphology for the SEM analysis [34,35]. All polypeptide samples after lyophilization showed three-dimensional (3D) networks with porous structures (Figure 2). Moreover, 24-armed samples exhibited larger pore sizes than 12-armed ones, which could facilitate the 24-armed samples to accommodate more water in the networks with lower CGCs [21,22].

X-ray diffraction (XRD) analysis was operated on lyophilized polypeptide samples with a range of 2θ = 5°–40° to characterize the molecular packing in the polypeptide hydrogel networks. As shown in Figure 3, all lyophilized polypeptide samples exhibited two major peaks at 2θ = 10°–12.5° and 20°–22.5°, which indicated the existence of the aromatic stacking by the grafted moieties and the amorphous state, respectively. Moreover, XRD data was used to compute the corresponding d spacing of the aromatic stacking ranged between 0.70 and 0.77 mm. [36,37].

The self-assembled nanostructures of the polypeptide hydrogels (8.0 wt%) were characterized by small-angle X-ray scattering (SAXS) analysis (Figure 4). For all hydrogel samples, the scattering intensity (*I*) exhibited a relation of *I*(*q*) ∝ *q*^−n^ with n = 3.2–3.8, indicating the network forming 3D assembled nanostructures with ill-defined morphology [38,39]. The SAXS data were fitted by SasView software to compute the radius of gyration (R_g_) of nano-assemblies formed by the packing of the grafted moieties. As shown in Appendix A, it is obvious that the 24-armed polypeptide hydrogels exhibited higher R_g_ than the 12-armed ones. The result showed that the polypeptide samples with the higher arm number had a tendency to form larger nano-assemblies in the hydrogel network.

### 2.5. Mechanical Strength and Recovery Behavior of Star-Shaped Graft Copolypeptides

The mechanical strengths of the polypeptide hydrogels (concentration: 8.0 wt%) were characterized by employing rheological measurements at room temperature (RT) (Figure 5). The strength properties of the samples were observed under two experimental conditions: changing of strain and angular frequency. The tendency of storage modulus (G’) and loss modulus (G”) following the experiments were considered to estimate the mechanical properties of the polypeptide hydrogels. It was obvious that all hydrogel samples showed the G’ values higher than G” regardless at various frequency (Figure 5a), showing no occurrence of sol-to-gel transition upon varying frequency. The 24-armed polypeptide hydrogels exhibited a higher mechanical strength than the 12-armed ones. As reported in the previous studies, the mechanical properties enhanced with the increase in polypeptide arm number [21,22,23,27]. The hydrogels formed by the polypeptides grafted with indole group exhibited higher mechanical properties than those formed by the polypeptides grafted with phenyl group with the same arm number. Previous studies have shown that the hydrogels formed by the polypeptides adopting more β-motifs exhibited higher mechanical properties [21,22,27,40,41]. Amongst all the hydrogel samples, 24s-PLL_12_-*g*-Indo_0.11_ adopted the highest percentage of β-motif conformation possessed the highest mechanical strength. As shown in Figure 5b, all the polypeptide hydrogel samples possessed shear-thinning behavior, evidenced by the decrease in G’ and G” upon the increase in strain. It could be seen a crossover point of G’ and G” at the strain lower than 100.0% for each sample, showing the strain limitation to deform the polypeptide hydrogel network. Moreover, the 24-armed polypeptide hydrogels exhibited crossover points at larger strain amplitude than the 12-armed ones, which indicated that the rigidity of the 24-armed polypeptide hydrogels were higher than that of 12-armed ones. The results showed that the mechanical strengths of polypeptide hydrogels were strongly dependent on grafted moiety and polypeptide topology.

To characterize the recovery behavior of the polypeptide hydrogels, the rheological measurements were operated at RT with the following conditions: fixed-frequency = 1 rad/s, various strain sweep values (1.0% for 100 s, 100.0% for 300 s, and 1.0% for 600 s). As shown in Appendix A, the G’ of hydrogel samples declined sharply as the strain increased to 100.0%, which is an indication of the breaking down of the gel structures, and the G’ value recovered to the original state after returning the strain to the initial value. The results showed that the polypeptide hydrogels exhibited good recovery ability, demonstrating the potential to be used as injectable materials [23,28].

## 3. Conclusions

In this study, we have reported the successful synthesis of the star-shaped poly(_L_-lysine) grafted with indole and phenyl groups using PGDs as the asymmetric initiators with various arm numbers. The 12-armed and 24-armed graft copolypeptides exhibited good hydrogelation ability with CGCs from 2.9 wt% to 6.3 wt% whilst the 6-armed counterparts could not undergo hydrogelation at a polypeptide concentration lower than 10.0 wt%. The polypeptide topology (arm number and symmetry) and various non-covalent interactions exerted by the grafted moiety played a critical role in the hydrogelation of the polypeptides. The packing of the grafted moiety resulted in the confinement of the PLL segment, facilitating the polypeptide chains to adopt more ordered conformation as well as hydrogelation. This study has given a potential insight for the synthesis and mechanical characteristic modification of the star-shaped polypeptide hydrogels by changing the polypeptide composition and network topology.

## 4. Materials and Methods

### 4.1. Materials

All chemicals were purchased and used directly without purification unless otherwise noted. ZLL NCA was synthesized by using the reported procedure [21,22,23,42]. PGD G1 (*M_n_* = 365), G2(*M_n_* = 800), and G3 (*M_n_* = 1690) were synthesized according to our previous reports [43]. Anhydrous hexane (ECHO, Tainan, Taiwan) and tetrahydrofuran (THF, J.T.Baker, Phillipsburg, NJ, USA) were used after the removal of water by stirring with calcium hydride (90–95%, Alfa Aesar, Lancashire, England) and sodium (99.95%, in mineral oil, Aldrich, Auvergne Rhone Alpes, France) overnight, respectively. DMF (Macron, Croydon, PA, USA) was dried by activated 4A molecular sieves (UniRegion Bio-Tech, Hsinchu, Taiwan).

### 4.2. Synthesis and Characterization of Star-Shaped Poly(Z-_L_-lysine) Polypeptides (s-PZLL)

Following the reported procedure, s-PLL homopolypeptides were synthesized by using ROP as shown in Figure 1 [21,22,27]. The reaction was promoted by using TMG as an activator for the OH groups on the initiators for ROP of ZLL NCA. In this study, PGDs G1, G2, and G3 were used as 6-armed, 12-armed, and 24-armed initiators for the synthesis of star-shaped polypeptides with an asymmetric topology, respectively. The synthesis of 6-armed poly(Z-_L_-lysine) (6s-PZLL) is given below as a detailed example.

Under an inert atmosphere, 11.65 mg of PGD G1 was dissolved in anhydrous DMF (1.634 mL) contained in a 60 mL round bottom flask with the addition of TMG (7.42 μL, Aldrich, Beijing, China) as a promoter. The solution was stirred and heated for 30 min until the mixture was becoming homogeneous. Then, 1.2 g of ZLL NCA was totally dissolved in anhydrous DMF (3.92 mL) in another bottle before dropping into the initiator solution. The flask with the reacting solution was sealed and stirred at RT for 48 h. Then, the solution was poured into a cellulose dialysis tube (MWCO: 6000–8000 g mol^−1^, Spectrum Laboratories, Rancho Dominguez, CA, USA) and dialyzed against methanol and DIW for 48 and 24 h, respectively. Finally, the dialyzed solution was lyophilized for 72 h to collect the white product (yield: 80–90%).

Each s-PZLL homopolypeptide sample was separately dissolved in TFA-*d*_1_ and DMF for ^1^H NMR and GPC-LS analyses, respectively, for polypeptide characterization. In this study, BRUKER ADVANCE III HD NMR (600 MHz, Bruker Corporation, Karlsruhe, Germany) system was employed to conduct ^1^H NMR at RT. For GPC-LS, a system with three Viscotek detectors and two Shodex columns was used to operate the analysis at 55 °C with 0.8 mL/min of flow rate. Moreover, 0.1 M LiBr in DMF solution and polystyrene (molecular weight: 25,000 g/mol) were requested as an eluent and a calculation standard, respectively, for this operation. After dissolving in DMF, the solutions used for GPC-LS analysis were passed through a filter (polytetrafluoroethylene, 0.45 mm, 13 mm, Finetech, Taichung, Taiwan) to remove the impurities.

### 4.3. Deprotection of Z-Group on Polypeptides

To synthesize star-shaped poly(_L_-lysine) homopolypeptides (s-PLL), the Z groups were removed by using hydrogen bromide (HBr, 33.0 wt% in acetic acid, Acros, Yehud, Israel) (Figure 1a) [21,22,27]. In detail, each 1.0 g of s-PZLL was dissolved in 50 mL of trifluoroacetic acid (TFA, Alfa Aesar, Lancashire, England) before HBr was dropped into the mixture slowly (molar ratio of HBr:Z group = 5:1). The reaction occurred at RT during stirring for 1 h. After completion of reaction, the solution was poured into diethyl ether for precipitation and the solvent was removed by centrifugation. The product was washed with diethyl ether twice and dried completely under vacuum overnight. The dried solid was dissolved in DIW and dialyzed against DIW for 72 h using a cellulose membrane with MWCO of 6000–8000 g mol^−1^. Lyophilization was employed for the removal of water to obtain the final product with a 90.0–97.0% of yield. The polypeptide sample after deprotection was dissolved in D_2_O for ^1^H NMR-600NMR analysis.

### 4.4. Synthesis and Characterization of Star-Shaped Poly(_L_-lysine) Graft Copolypeptides

For functionalization of star-shaped polypeptide, the amino group on s-PLL was reacted with benzoic acid and 3-indoleacetic acid using EDC/NHS as activators (Figure 1a) [23,28]. The synthesis of 6-armed poly(_L_-lysine)-*graft*-indole (6s-PLL-*g*-Indo) is shown as a reference.

In the sequence, EDC (0.56 mL, Alfa Aesar, Lancashire, England) and NHS (0.368 g, Alfa Aesar, Lancashire, England) were added to the mixture of 3-indoleacetic acid (83.96 mg) dissolved in anhydrous methanol (2.1 mL, AENCORE, Box Hill, Australia) under a nitrogen atmosphere. The solution was sealed and stirred at RT for 24 h for EDC/NHS to activate the carboxyl groups on 3-indoleacetic acid (Sigma-Aldrich, St. Louis, MO, USA). The 6s-PLL polypeptide (425 mg) was dissolved in anhydrous methanol (4.25 mL) before being added to the activated solution. The final solution was stirred for 48 h, dialyzed against DI water for 48 h with a dialysis membrane (MWCO of 6000–8000 g mol^−1^), and lyophilized for 72 h to obtain a spongy product (yield: 90.0–95.0%). The graft copolypeptide was dissolved in D_2_O for ^1^H NMR-600NMR analysis.

### 4.5. Preparation of Star-Shaped Graft Copolypeptide Hydrogel Samples and Determination of Critical Gelation Concentrations (CGCs)

Star-shaped graft copolypeptide samples were dissolved in DIW with various concentrations to analyze their hydrogelation ability with the assistance of vortex and sonication to obtain homogeneous mixtures. To determine the CGCs of the samples, the inverting vial method was employed to observe whether the polypeptide mixtures will flow down after the containers were reversed [44,45,46]. The sample exhibit hydrogelation ability if it is stable at the vial bottom after 60 s with a concentration lower than 10.0 wt%. CGC is the lowest concentration of polypeptide at which it could undergo hydrogelation.

### 4.6. Determination of Secondary Structure Adoption

CD analysis was employed to characterize the polypeptide secondary conformation. The polypeptide samples dissolved in DIW at a concentration of 0.1 mg/mL were analyzed by CD measurement using a JASCO J-815 spectrometer (JASCO Corporation, Tokyo, Japan) at RT. The percentages of secondary conformations were calculated from CD data by using BeStSel fitting website.

### 4.7. Characterization of Graft Copolypeptide Hydrogel Self-Assembled Nanostructures

The self-assembled nanostructures of hydrogel samples were determined by using XRD, HR-SEM, and SAXS analyses. The 12-armed and 24-armed polypeptide hydrogels were prepared at concentrations of 8.0 and 5.0 wt%, respectively, and lyophilized for XRD and SEM measurements. Ultima IV-9407F701 Model of X-ray spectrometer (Rigaku Corporation, Tokyo, Japan) equipped with Cu Kα radiation (250 mA, 50 kV, 0.154 nm) at 4°/min of speed was used to determine XRD data of freeze-dried samples from 2θ = 5°–40°. The lyophilized samples were sputter-coated with platinum (Pt) before the operation of SEM imaging by using A Hitachi SU8010 microscope (Hitachi High-Technologies Corporation, Tokyo, Japan). For SAXS analysis, the functioned copolypeptide hydrogel samples were prepared at a concentration of 8.0 wt%, contained in quartz capillary tubes (1 mm diameter), and analyzed by a NANOSTAR U SYSTEM (Bruker AXS Gmbh, Karlsruhe, Germany) under 650 μA of current and 45 kV of voltage at RT.

### 4.8. Characterization of Mechanical Strength and Recovery Behavior

The mechanical strength and recovery behavior of hydrogel samples were characterized by the storage modulus (G’) and loss modulus (G”) values of rheological measurements. The hydrogel samples were prepared at a concentration of 8.0 wt% for all conditions of rheological measurements at RT. The mechanical properties of samples were studied under various strain sweep (frequency = 1 rad/s, strain = 0.01–100.0%) and frequency (strain = 1.0%, frequency = 0.1–100 rad/s) conditions. To observe the recovery ability of the hydrogel samples, the rheological system was operated at a fixed frequency of 1 rad/s and changing strain sweep under three different periods: 1.0% for the first 100 s, 100.0% for the next 300 s to break the hydrogel structure down, and 1.0% for the last 600 s. A rheometer HR-2 system (TA Instruments/Waters Corporation, New Castle, DE, USA) was employed to measure the rheological properties in this study, with a parallel aluminum plate of 25 mm diameter.

## Data Availability

Not applicable.

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
