# Peer review of "Synthesis and Hydrogelation of Star-Shaped Graft Copolypetides with Asymmetric Topology"

_gels, 2022, doi:10.3390/gels8060366_

Round 1
Reviewer 1 Report
The novelty of the work must be better described in the introductory part; The importance of tuning gel characteristics by changing its composition should be described better and related to specific applications;
As applications envisaged the biological ones, what authors can comment on the cytotoxicity of their gels? Can they introduce at least some preliminary data on biocompatibility with a mammalian cell line?
Author Response
Reviewer 1:
The novelty of the work must be better described in the introductory part; The importance of tuning gel characteristics by changing its composition should be described better and related to specific applications;
Response: The authors thank the reviewer’s comments. We have described and explained the novelty of our research in lines 64-70 of page 2.
“One previous paper has used PGD G2 as an initiator for the synthesis of the block copolypeptides and studied the hydrogelation of these polypeptides [21]. The PGD G1 and G3 have never been used as initiators for the synthesis of polypeptides. Herein, this study reported the synthesis and hydrogelation of asymmetric, star-shaped graft copolypeptides with 6, 12 and 24 arms using PGD G1, G2 and G3 as initiators, respectively. Moreover, indole and phenyl groups were employed as the grafting moieties for the polypeptides.”
As applications envisaged the biological ones, what authors can comment on the cytotoxicity of their gels? Can they introduce at least some preliminary data on biocompatibility with a mammalian cell line?
Response: The authors thank the reviewer’s comments. In this study, we have focused on the synthesis and hydrogelation of polypeptides without investigating their applications, especially in the biological fields. However, in 2009, Sofroniew’s group reported the application of amphiphilic diblock copolypeptide hydrogels in the biological field [Biomaterials 30 (2009) 2881–2898]. This study demonstrated that polylysine-block-polyleucine copolypeptide hydrogels had no detectable toxicity to neurons, myelin, and axons. Moreover, our group has done two reports working on the applications of the graft copolypeptide hydrogels and chemically crosslinked polypeptide nanogels in 2019 [Nanoscale, 2019, 11, 11696; Materials Science & Engineering C 102 (2019) 85-95]. Both of them showed that polypeptides had negligible toxicity. Currently, we are investigating the anticancer activity of these star-shaped polypeptides.
Reviewer 2 Report
This paper described the synthesis of the star-shaped polypeptides grafted with indole and phenyl groups, using polyglycerol dendrimers (PGDs) G1, G2, and G3 as the asym metric initiators to investigate the effect of grafted moiety, arm number, and polypeptide topology on the hydrogelation of the star-shaped graft copolypeptides. The 12-armed and 24-armed graft copolypeptides exhibited good hydrogelation ability with CGCs from 2.9 wt% to 6.3 wt% whilst the 6-armed counterparts could not undergo hydrogelation at a polypeptide concentration lower than 10.0 wt%. The polypeptide topology (arm number and symmetry) and various non-covalent interactions exerted by the grafted moiety played a critical role in the hydrogelation of the polypeptides. The packing of the grafted moiety resulted in the confinement of the PLL segment, facilitating the polypeptide chains to adopt more ordered conformation as well as hydrogelation. The article was well done and brings this new methodology to synthesize star-shaped graft copolypeptides. It is a basic article and only showed the characterization step.
Author Response
Reviewer 2:
This paper described the synthesis of the star-shaped polypeptides grafted with indole and phenyl groups, using polyglycerol dendrimers (PGDs) G1, G2, and G3 as the asymmetric initiators to investigate the effect of grafted moiety, arm number, and polypeptide topology on the hydrogelation of the star-shaped graft copolypeptides. The 12-armed and 24-armed graft copolypeptides exhibited good hydrogelation ability with CGCs from 2.9 wt% to 6.3 wt% whilst the 6-armed counterparts could not undergo hydrogelation at a polypeptide concentration lower than 10.0 wt%. The polypeptide topology (arm number and symmetry) and various non-covalent interactions exerted by the grafted moiety played a critical role in the hydrogelation of the polypeptides. The packing of the grafted moiety resulted in the confinement of the PLL segment, facilitating the polypeptide chains to adopt more ordered conformation as well as hydrogelation. The article was well done and brings this new methodology to synthesize star-shaped graft copolypeptides. It is a basic article and only showed the characterization step.
Response: The authors thank the reviewer’s comments.

Reviewer 3 Report
This study is set to synthesize the star-shaped poly (L-lysine) grafted with indole and phenyl groups using polyglycerol dimers as the asymmetric initiators with various arm numbers that have potential as biomaterials e.g., wound dressing, drug delivery systems, and tissue engineering. They have successfully demonstrated that the star-shaped polypeptides grafted with indole group on the side chain had better hydrogelation ability than those with phenyl group with the same arm number. Synthesis and characterization of star-shaped graft copolypeptides, critical gelation concentrations, the secondary structure of star-shaped graft copolypeptides, and mechanical strength and recovery behavior of star-shaped graft copolypeptides have been successfully determined. Results obtained are presented and discussed well in the light of the previous related studies. As a result, the data obtained from this study are sound and proved to be useful for various medical applications. Therefore, I have concluded that this study deserved to be published in the journal Gels. Moreover, some minor corrections need to be done before the publication of this manuscript, as given below:
1. Figure S1: Proton (b) is not visible in the spectra. Also, (a) and (c) looks quite similar. Authors are suggested to include original well-integrated NMR spectra of the compounds. Include 13C NMR of the dendrimers.
2. Fig. 1c: Methylene protons (CH2, proton labeled as c) are not assigned.
3. Abstract: Authors stated, “The packing of the grafting moiety via non-covalent interactions not only facilitated the polypeptides…” However, I do not see any discussion on NCI in the manuscript.
4. Page 2, line 91: What is free drying?
5. Scheme 1: Define the abbreviations.
6. Line 220: polypeptide is misspelled as “polypetide”.
Author Response
Reviewer 3:
This study is set to synthesize the star-shaped poly (L-lysine) grafted with indole and phenyl groups using polyglycerol dimers as the asymmetric initiators with various arm numbers that have potential as biomaterials e.g., wound dressing, drug delivery systems, and tissue engineering. They have successfully demonstrated that the star-shaped polypeptides grafted with indole group on the side chain had better hydrogelation ability than those with phenyl group with the same arm number. Synthesis and characterization of star-shaped graft copolypeptides, critical gelation concentrations, the secondary structure of star-shaped graft copolypeptides, and mechanical strength and recovery behavior of star-shaped graft copolypeptides have been successfully determined. Results obtained are presented and discussed well in the light of the previous related studies. As a result, the data obtained from this study are sound and proved to be useful for various medical applications.
Response: The authors thank the reviewer’s comments above and revised our manuscript according to the specific comments.
Therefore, I have concluded that this study deserved to be published in the journal Gels. Moreover, some minor corrections need to be done before the publication of this manuscript, as given below:
- Figure S1: Proton (b) is not visible in the spectra. Also, (a) and (c) looks quite similar. Authors are suggested to include original well-integrated NMR spectra of the compounds. Include 13C NMR of the dendrimers.
Response: The authors thank the reviewer’s comment. We have enlarged the proton b to be obvious in Figure S1 (Supporting Information).
We have shown 1H NMR spectra of s-PZLL samples with integrated values below. We have published the 13C NMR spectra of the dendrimers in our previous papers [Langmuir 2021, 37, 8534-8543; Polymer 250 (2022) 124864].
Figure 1. 1H NMR spectra of 6s-PZLL32 in TFA-d1.
Figure 2. 1H NMR spectra of 12s-PZLL12 in TFA-d1.
Figure 3. 1H NMR spectra of 24s-PZLL12 in TFA-d1.
- 1c: Methylene protons (CH2, proton labeled as c) are not assigned.
Response: The authors apologize this omission. We have assigned the proton c in Figure 1c of page 6.
- Abstract: Authors stated, “The packing of the grafting moiety via non-covalent interactions not only facilitated the polypeptides…” However, I do not see any discussion on NCI in the manuscript.
Response: The authors thank the reviewer’s comment. Non-covalent interactions include aromatic and hydrophobic effects. We have mentioned their effects on hydrogelation of polypeptide samples in lines 152-158 of pages 7-8 as the paragraph below.
“The hydrogen bonding, hydrophobic, and aromatic (including cation-p, p-p) interactions of the indole group could establish efficient facilitation for the hydrogelation of the star-shaped polypeptides. Although the grafted phenyl group cannot efficiently facilitate the formation of hydrogels as good as the grafted indole group, the hydrophobic and aromatic interactions of the phenyl group still could assist the hydrogelation of the corresponding polypeptides with low CGCs (6.3 wt% for 12s-PLL12-g-Phenyl0.08 and 3.6 wt% for 24s-PLL12-g-Phenyl0.13).”
- Page 2, line 91: What is free drying?
Response: The authors apologize for this mistake. We have corrected “free-drying” to “freeze-drying” in line 97 of page 2.
- Scheme 1: Define the abbreviations.
Response: The authors thank the reviewer’s comment. We have added the abbreviations in the title of Scheme 1 in lines 103-107 of page 3.
- Line 220: polypeptide is misspelled as “polypetide”.
Response: The authors apologize for this mistake. We have corrected “polypetide” to “polypeptide” in line 230 of page 10.
